# The Greek Collaborative Long COVID Study: Non-Hospitalized and Hospitalized Patients Share Similar Symptom Patterns

**DOI:** 10.3390/jpm12060987

**Published:** 2022-06-17

**Authors:** Martha-Spyridoula Katsarou, Eleni Iasonidou, Alexander Osarogue, Efthymios Kalafatis, Maria Stefanatou, Sofia Pappa, Stylianos Gatzonis, Anastasia Verentzioti, Pantelis Gounopoulos, Christos Demponeras, Eleni Konstantinidou, Nikolaos Drakoulis, Andreas Asimakos, Archontoula Antonoglou, Aspasia Mavronasou, Stavroula Spetsioti, Anastasia Kotanidou, Paraskevi Katsaounou

**Affiliations:** 1Research Group of Clinical Pharmacology and Pharmacogenomics, Faculty of Pharmacy, School of Health Sciences, National and Kapodistrian University of Athens, 15771 Zografou, Greece; mskatsarou@gmail.com (M.-S.K.); drakoulis@pharm.uoa.gr (N.D.); 2Long COVID Greece Patient Society, 57010 Thessaloniki, Greece; iasonidouhelen@hotmail.com (E.I.); themos.kalafatis@gmail.com (E.K.); 3Medical School, National and Kapodistrian University of Athens, 15784 Athens, Greece; marwstef@gmail.com (M.S.); sgatzon@med.uoa.gr (S.G.); nverentzioti@gmail.com (A.V.); silverakos@gmail.com (A.A.); arxoantonoglou@gmail.com (A.A.); aspasiamavronasou@hotmail.com (A.M.); roula_spe@hotmail.com (S.S.); akotanid@med.uoa.gr (A.K.); 4Institute for Social and Economic Sciences, School of Economics E.H.E. Europa Hochschule EurAka, 4106 Therwil, Switzerland; a.osarogue@gmail.com; 51st Department of Neurosurgery, “Evangelismos” Hospital, National and Kapodistrian University of Athens, 10676 Athens, Greece; 6Department of Brain Sciences, Imperial College London, London W6 8RF, UK; sofia.pappa@westlondon.nhs.uk; 72nd Cardiology Department, “Evangelismos” Hospital, 10676 Athens, Greece; gounopoulos@yahoo.gr (P.G.); el.kon@outlook.com.gr (E.K.); 8General Intensive Care Unit, Thoracic Diseases General Hospital Sotiria, 11527 Athens, Greece; ch_demponeras@hotmail.com; 9Respiratory Department First ICU, Evangelismos Hospital, 10676 Athens, Greece; 10Clinical Exercise Physiology and Rehabilitation Laboratory, Physiotherapy Department, University of Thessaly–Lamia, 38221 Lamia, Greece

**Keywords:** Long COVID, post-COVID, post-acute COVID (PACS), post-acute sequelae of COVID-19 (PASC)

## Abstract

Long COVID-19 syndrome refers to persisting symptoms (>12 weeks) after the initial coronavirus infection and is estimated to affect 3% to 12% of people diagnosed with the disease globally. Aim: We conducted a collaborative study with the Long COVID patient organization in Greece, in order to estimate the characteristics, symptoms, and challenges these patients confront. Methods: Data were collected from 208 patients using unstructured qualitative free-text entries in an anonymized online questionnaire. Results: The majority of respondents (68.8%) were not hospitalized and reported lingering symptoms (66.8%) for more than six months. Eighteen different symptoms (fatigue, palpitations, shortness of breath, parosmia, etc.) were mentioned in both hospitalized and community patients. Awareness of Long COVID sequelae seems to be low even among medical doctors. Treatment options incorporating targeted rehabilitation programs are either not available or still not included inthe management plan of Long COVID patients. Conclusions: Patients infected with coronavirus with initial mild symptoms suffer from the same persistent symptoms as those who were hospitalized. Long COVID syndrome appears to be a multi-systemic entity and a multidisciplinary medical approach should be adopted in order to correctly diagnose and successfully manage these patients.

## 1. Introduction

The long-term effects of SARS-CoV-2 (COVID-19) area major concern for stakeholders in the health community as the pandemic has not only caused extraordinary morbidity, mortality, and global disruption [1,2] but also seriously affected the recovery of a number of COVID-19 survivors [1].

It is now well established that SARS-CoV-2 causes systemic infection and may have long-lasting consequences for weeks or even months after diagnosis [3].A variety of terms can be found in the literature to date, including Post-acute Sequelae of COVID-19 (PASC), Long COVID, Post-acute COVID-19 Syndrome (PACS), Chronic COVID-19, and Long Haul COVID-19; each term has a slightly varying definition [1]. The term ‘Chronic COVID’ has been used to describe symptoms lasting more than 12 weeks, whereas the term ‘Post-acute COVID’ refers to persistent symptoms 3 weeks after COVID-19 infection. The Center for Disease Control (CDC) adopted a description for Long COVID to include new or persistent symptoms at 4 or more weeks from infection with SARS-CoV-2 [4].

The term Long COVID has been conceptually used to identify the presence of any symptoms after COVID-19 infection and consists of two stages: (1) post-acute consequences of SARS-CoV-2 infection (PASC) or acute post-COVID infection (from week 5 to week 12 after the onset of symptoms) and (2) Long COVID (for symptoms lasting more than 12 weeks). Moreover, the recurrence of features after COVID-19 diagnosisshould be recorded, as it remains pertinent to determine the pattern (variations) and nature (new onset or worsening) of any symptoms [5].

An estimated 3% to 12% of people infected with coronavirus have symptoms 12 weeks after the initial infection [6]. The incidence and evolution of PASC seem to be dependent on the duration of the initial infection, systems affected, variant of the virus, severity of acute phase, and geographic region [7]. Harmonizing research for improved understanding of the novel condition has led to more comprehensive definitions, which incorporate the key effects on both adults and children [8,9]. Prevalence estimates for Long COVID, specifically in children and young people, vary from 1% to 51%. The CLoCk Study estimated that the rates of non-hospitalized children and young people with Long COVID were up to 14% [8]. Reducing the variation in the description is paramount in order to establish a common understanding of the prevalence of specific symptoms after COVID-19 [2]. In our manuscript, we adopted the CDC’s definition of Long COVID (symptoms at 4 or more weeks from infection with SARS-CoV-2).

Although more than 200 symptoms have been associated with Long COVID to date, only common symptoms include cough, shortness of breath, fatigue, headaches, anosmia, and general lack of wellbeing, according to the National Institute of Health [10], havean impact on everyday functioning [11]. However, Chen et al. using corresponding estimated pooled symptom-specific prevalence obtained from the 23 symptoms assessed across 30 studies, pinpoint the five most prevalent symptoms as fatigue, dyspnea, insomnia, joint pain, and memory problems [2]. Conversely, Groff et al. underline that the most frequently observed or reported symptoms involve functional mobility impairments, pulmonary abnormalities, and mental health disorders [7]. Other symptoms include impaired vision, red eyes, vertigo, muscle pain, loss of appetite, diarrhea, skin lesions, joint pains, headache, rhinitis, loss of smell, Sjogren’s syndrome, loss of taste, sore throat, chest pain, and cough.

COVID-19 is a multisystem disease, though its pathophysiology is still poorly understood [12]. SARS-CoV-2 infects the host using the angiotensin-converting enzyme 2 receptors, which are expressed in several organs, including the lung, heart, kidney, intestine, and endothelial cells, causing a distinguishable and distinct systemic endotheliitis [12,13]. Various mechanisms have been proposed in the literature on Long COVID effects such as systemic inflammation, involvement of the nervous system, autonomic neuropathy, the effects of hypoxia and hypovolemia, endothelitis, the effects of the cytokine cascade, post-ICU syndrome, the remaining of the virus in the body, and a late immune response. In the study by Charfeddine et al., persistent symptoms, especially chest pain, fatigue, and neurocognitive symptoms were mainly associated with endothelial dysfunction [14]. Weschler et al. highlighted the potential role of mast cell activation in a subgroup of patients with PASC [15]. Phetsouphanh et al. implicated immune dysregulation given that Long COVID patients had highly activated innate immune cells, lacked naive T and B cells, and showed elevated expression of type I IFN (IFN-β) and type III IFN (IFN-λ1) that remained persistently high 8 months after infection [16]. The patterns may also be explained by autonomic instability and may result from deconditioning, hypovolemia, or immune- or virus-mediated neuropathy [1]. Chen et al. believe that Long COVID conditionsmay be related to a virus- or immune-mediated disruption of the autonomic nervous system resulting in orthostatic intolerance syndromes [2].Furthermore, these mechanisms can be grouped into the direct effect of the viral infection and the indirect effect on mental health due to posttraumatic stress, social isolation, and economic factors, such as loss of employment [14,15,17,18].

In addition, the Mount Sinai algorithm highlights the importance of establishing the absence of other medical causes before attributing persistent or emerging symptoms to PASC or Long COVIDsyndrome as seen in Figure 1.

The current study was co-produced in collaboration with the Long COVID Organization in Greece and aims primarily to provide an insight into the prevalence, demographic data, and symptoms of Long COVID patients in Greece. Other aspects such as awareness, the economic burden of the disease, impact on employment and degree of occupational support as well as the level of utilization of available resources have also been explored as secondary outcomes.

## 2. Materials and Methods

An online questionnaire was distributed to a cohort of Long COVID patients who are members of the Long COVID Greece patients’ society onFacebook as shown in the flow chart below (Figure 2). Only adults 18 years and above were included in the survey.

The survey was carried out using forms specifically designed to collect data with anonymized IP and the participants were informed about the anonymization procedure. Symptoms were descriptive as patients used free-text entries. The free text has been analyzed by identifying term frequencies and common bigrams (sequence of two adjacent terms) and trigrams (sequence of three adjacent terms).

Incorrect entries and misspellings have been identified using the Levenshtein distance algorithm [19] (with distance setting = 1). Subsequently, we used Information Extraction to automatically identify symptom entries and generate true/false features in the case that a specific symptom was either present or absent on a patient level, in order to minimize the time required for feature generation. These entries were then checked for their validity by two investigators and corrected where applicable. The use of the decision tree model has been handy in the precise classification of patients into specific groups based on demographics, time of disease, hospitalization, and expressed symptoms. For the data hypothesis, a decision tree algorithm (Knime Framework) has been used [20,21,22]. The method used is “Decision Tree Learner”, using “Gini” to split data [23] “class column” and “minimum number entries per node” was set to 30. Since we focus oncreating hypotheses rather than evaluating the predictive capabilities of our algorithm, no train, test, or validation data subsets were used, and the decision tree analysis was performed using our entire dataset. Interactions between symptoms were revealed with network analysis [24].

Comparison analysis between symptoms of hospitalized and non-hospitalized patients was performed using the Chi-Square X2 test, contingency tables 2 × 2 (significance level of a = 0.01), and a statistical significance *p*-value was calculated on a free platform https://www.socscistatistics.com/tests/chisquare2/default2.aspx, (accessed on 1 February 2022).

The English translation of the questionnaire used is seen in Table 1.

## 3. Results

The sample includes 208 patients (77.9% female/male 22.1%). The majority of respondents fall within the age range of 41–50 years. The duration since diagnosis with Long COVID was <1 month (3.9%), 7 to 12 months (44.7%), 1 to 6 months (29.3%), >12 months (22.1%). Regarding hospitalizations and admissions, 68.8% have not been hospitalized, 26.9% have been hospitalized outside ICU, while 1.3% required admission toan ICU. Among non-hospitalized patients, 69.44% still suffered from fatigue and 15.27% from dyspnea, 28.7% from difficulty in concentration, and 28.4% from tachycardia.

Regarding awareness and utilization of Long COVID rehabilitation resources, 26.9% of patients had already completed post-COVID-19 rehabilitation, while 22.1% wereoblivious of suchprograms. Regarding access to Health Care System, 78.4% of patients preferred tertiary hospitals, while 21.6% preferred primary healthcare institutions.

### 3.1. Persistent Symptoms

Eighteen different symptoms were most frequently mentioned by the Greek group of Long COVID haulers (Figure 3).

Despite the variation in the initial severity of acute phase, patients with initial mild symptoms suffer from the same persistent symptoms as hospitalized. No significant differences were found in most symptoms between the two groups (hospitalized and non-hospitalized) as shown in Table 2.

#### 3.1.1. Fatigue

Fatigue is a common symptom in both groups and in fact the most common of all symptoms in our cohort.

#### 3.1.2. Musculoskeletal

Musculoskeletal symptoms were present in both groups but more common in the hospitalized subgroup, without reaching any statistical difference.

#### 3.1.3. Neurological, Cognitive, and Psychiatric Symptoms

The majority of symptoms described seem to be related to the nervous system. Specifically, 75% of participants had at least one neurological symptom several months after infection such as headache, dysautonomia, muscle aches, muscle spasms, dizziness, tinnitus, and cognitive disorders (lack of concentration, memory disturbances, and brain fog). Additionally, 70% of patients reported fatigue, probably related to nervous system dysfunction. Mood disorders such as anxiety and depression were 6.5 times more common in hospitalized patients, reflecting a statistically significant difference between the two groups.

Interestingly, parosmia and other olfactory disorders were presented mostly in the mild acute phase group of patients (Table 1).

#### 3.1.4. Palpitations

Palpitations are found in more than one quarter of both Long COVID patient groups.

#### 3.1.5. Shortness of Breath

Shortness of breath is found in 23.43% of the hospitalized patients in comparison to 15.27% of the non-hospitalized patients.

#### 3.1.6. Dermatological

In relation to dermatological symptoms, the most commonly reported symptom was prolonged hair loss. Other dermatological disturbances like acne, psoriasis, and rashes have also been reported in this study group.

### 3.2. Socioeconomic Impact

Regarding the impact of Long COVID on employment, about 58.2% of patients are faced with work difficulties due to prolonged symptoms. Furthermore, when asked about the level of understanding of the disease in their work environment, 29.8% responded positively, while 20.2% of the patients seem to deal with a negative attitude.

Moreover, 12% of the respondents are attended to by public hospitals while 88% are not. As shown in Figure 3, only 11% of our Long COVID patients have never visited a physician (20 persons) for their symptoms, but from the ones that are followed up by medical staff, 53.4% have not been able to establish an underlying medical condition that could be responsible for their symptoms. Namely,30.8% are seen by private doctors, 13.5% by a mix of public and private doctors, and 2.3% are by health professionals in public hospitals. With respect to the number of medical resources used while having the ailment, 11.5% affirmed they have not visited different doctors to resolve their symptoms, 42.3% have visited at most two different doctors, 42.3% have visited up to tendifferent doctors, while 3.9 % have visited more than tendifferent doctors to find a solution to symptoms. Similarly, when asked whether they’ve encountered doctors with awareness of the ailment, 46.6% responded having encountered doctors who are unaware of Long COVID, 25% responded in negation, and 28.4% believe doctors they encountered are aware but without any active engagement. Regarding health care utilization costs, 47.6% have spent at least 500 euro on medical care, 31.3% have spent over 500 euro while 21.2% have spent nothing. (Figure 4)

The decision tree analysis shows clearly that fatigue is a key factor which is positively related to employment issues. Patients who report occupational issues show higher rates of fatigue and visited a higher number of medical professionals in order to be diagnosed and treated (Figure 5).

## 4. Discussion

Following an initial period of distrust of patients who continued to complain of symptoms or develop new symptoms long after their initial COVID-19 infection, a number of symptoms and signs were identified as components of the Long COVID syndrome [11].

It is worth emphasizing that our study is one of the few comparing LongCovid symptoms in hospitalized and non-hospitalized patients; from our cohort, 68.8% of patients have not been hospitalized, while only 4.3% were admitted tothe ICU. However, the description of persisting symptoms is quite similar across all aforementioned categories.

As also observed in our sample, symptoms correlating to neurological and neuropsychiatric dysfunction are common, including headache, cognitive impairment, smell/taste disorders, sleep and mood disorders, myalgias, and dysautonomia, thereby representing core aspects of the Long COVID Syndrome [25]. Similarly, summarizing evidence to date, a recently published systematic review and meta-analysis, including data from 47,910 patients, showed that up to 80% of infected patients developed one or more long-lasting neuropsychiatric symptoms, including fatigue (58%), headache (44%), attention disorder (27%), ageusia (23%), anosmia (21%), memory loss (16%), anxiety (13%), and depression (12%) [26].

Furthermore, the severity of acute COVID-19 symptoms and the need for hospitalization and/or ICU admission [26] have been associated with an increased risk of developing Long COVID syndrome [27]. However, non-hospitalized or patients with milder forms of the illness have been also found to frequently develop neuropsychiatric sequelae. For example, a large retrospective cohort study reported that the incidence of neurological or psychiatric diagnoses at 6 months post-COVID-19 infection was 33%, with an increased risk even for patients who were not hospitalized [26].

In fact, there was no significant difference between the two groups (hospitalized vs. non-hospitalized) in the frequency of neurological complaints in our study with the notable exception of olfactory disorders. Parosmia and other smell disorders were most commonly reported in the non-hospitalized group of patients which is in accordance with previous reports showing that olfactory dysfunction appears to be a component of LongCOVID, with parosmia as a prominent symptom in almost half of those with smell loss [28]. In a recent study byPérez–González et al., extra-thoracic symptoms (39.1%), chest symptoms (27%), dyspnoea (20.6%) and fatigue (16.1%) were more common in hospitalized patients (52.3% vs. 38.2%) and in women (59.0% vs. 40.5%) [29]. In another recent study byAli et al., non-hospitalized COVID-19 patients continue to experience neurologic symptoms, fatigue, and compromised quality of life 14.8 months after initial infection [30]. This is in accordance with our study where 75% of participants had at least one neurological symptom several months after infection.

Furthermore, according to our studies, hospitalized patients had significantly more psychological complications than patients that did not require hospital admission. For example, mood symptoms were significantly more frequent in the hospitalized group and quite low in the rest of the sample. This is at odds with a recent study in predominantly community patients with the post-COVID syndrome in Brazil whereby 39.8% of individuals reported memory problems, 36.9% anxiety, 44.9% depression, and 45.8% sleep problems [31]. However, prevalence rates vary considerably across studies, and estimates for mental health symptoms appeared to be lower in COVID-19 survivors in Greece post-hospitalization (depression 19%, anxiety 27%, traumatic stress 39%, and insomnia 33%) [32]. Nevertheless, it is difficult to make inferences about the potential role of hospitalization and initial disease severity solely based on this observation though it is well established that the need for admission and a history of hypoxia and pharmacotherapy can affect the psychological status and response of patients.

There is now a consensus that Long COVID is a multi-systemic entity that requires multiple partners to care for patients. Moreover, a number of symptoms could be attributed to several systems, and therefore there is a need for a thorough differential diagnosis and a collaboration of multiple medical experts. Namely, shortness of breath is mainly attributed to respiratory disorders: Given that COVID-19 is a virus that primarily affects the respiratory system residual dyspnea is usually attributed by both patients and health care professionals to pneumonia or fibrosis caused by the infection. However, dyspnea could be caused additionally by cardiovascular disorders and deconditioning. Therefore, a careful differential diagnostic algorithm should be followed before the treatment of dyspneic Long COVID patients with CPET having a central role. Moreover, palpitations, which relateto tachycardia could be caused by hypoxemia, hyperthyroidism, cardiovascular disease, or neurological diseases such as Postural Orthostatic Tachycardia Syndrome (POTS) and dysautonomia. The identification of autonomic dysfunction following COVID-19 infection involves the careful evaluation of the individuals expressing breathlessness, palpitations, fatigue, chest pain, presyncope, or syncope [1]. In this aspect, we proposed a diagnostic algorithm by adapting already used algorithms with data from the Greek Long COVID cohort in order to target the Greek population and its characteristics (Figure 6).

Our proposal for a holistic interventional approach using a multidisciplinary medical team is in fact using personalized medicine for our Long COVID patients. However, its effectiveness in securely diagnosing the underlying cause of Long COVID symptoms remains to be proved.

Our proposal implies that involved physicians and other healthcare professionals are well-trained for the accurate recognition of such cases, appreciate the symptom burden, and provide supportive management. It is also important to note that most of our patients faced skepticism and disbelief from their families, professional environment, and even from health professionals because of their ignorance of post-COVID-19 symptoms. Therefore, both awareness campaigns for the public and educational programs for health care professionals should be carefully designed. Employing a clinical framework that ensures that all initial appointments are based on predominant symptoms, and evaluations and treatment are determined on a case-by-case basis, is pertinent. Furthermore, psychological and neuropsychiatric complications are commonly encountered and should be proactively screened for and differentiated from other overlapping physical complaints as well assertively managed in a multi-disciplinary manner.

Patient rehabilitation is essential to avert re-hospitalizations and other long-term complications due to the aforementioned complications and other Long COVID-related symptoms, as well as improve patients’ post-COVID quality of life.However, it is important to note that the seriousness of hospitalization seems to be a defining factor inpatients accessing rehabilitation in a hospital/clinical setting. Namely, almost all patients that were admitted to an ICU followed a rehabilitation program after being discharged. Onthe contrary, most of the non-hospitalized patients were not offered such a program and the majority were completely unaware of this possibility. 52% of patients reported having tried rehabilitation during and post disease.

Our study is the first study in Greece that records data for non-hospitalized Long COVID patients. However, it is difficult to establish any causative correlation between non-hospitalized and hospitalized patients due to the small sample size involved. In addition, symptoms are self-reported and there is no formal clinical screening involved.

A limitation of our study is that it has involved a population partially selected from aFacebook group under the name of “Long COVID Greece patients” whichcould have led to a selection of patients already more aware of the disease alongside a potential age selection bias against possibly older patients not familiar with Facebook, have not been included to the study.Another limitation of our study is that symptoms were self-reported. Nevertheless, it is worth mentioning that till now Long-COVID is globally defined through self-reported symptoms and often web is used for reporting these symptoms.In our study, we additionally tried to reduce the bias by using free-text recorded symptoms and included a population (89%) that has been examined by medical doctors.

Finally, given the fact that non-hospitalized and hospitalized Long COVID patients share similar symptom patterns, Long COVID may rather be genetically driven than acute phase severity dependent. It is well known that there are significant genetic variations even between Caucasian populations [33] and this may explain different outcomes in studies mentioned above [31,32]. Nevertheless, further research is necessary in order to reveal the nature of the post-COVID syndrome or syndromes.

## 5. Conclusions

To the best of our knowledge our study seems to be among the first studies on Long COVID to address non-hospitalized patients comparing their symptoms to those patients who had been previously hospitalized showing lingering symptoms for more than one year and similarities between the two groups, but with more psychological complications in the hospitalized ones. Moreover, it is the only study done in Greece and Eastern Europe throwing research light on the aforementioned. Additionally, socioeconomic outcomes of Long COVID have been addressed.

A personalized approach to studying Long COVID patients with larger sample studies with genetic analysis isneeded in order to understand the pathophysiology of the disease and devise effective treatments.

## Figures and Tables

**Figure 1 jpm-12-00987-f001:**
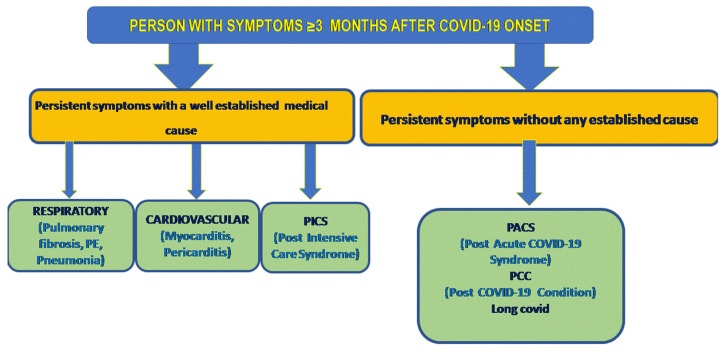
Adjusted algorithm from Mount Sinai 2020 (Tabacof-Rehabilitation management of autonomic dysregulation in Post COVID-19 Condition (who.int)).

**Figure 2 jpm-12-00987-f002:**
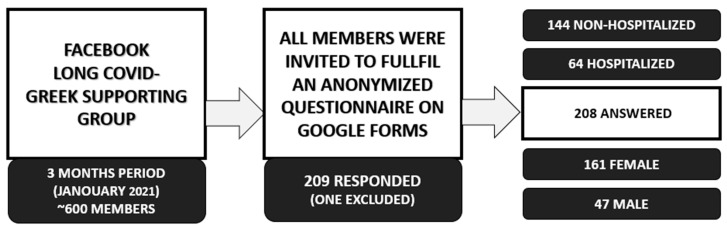
Long COVID Greek Supporting group response to questionnaire.

**Figure 3 jpm-12-00987-f003:**
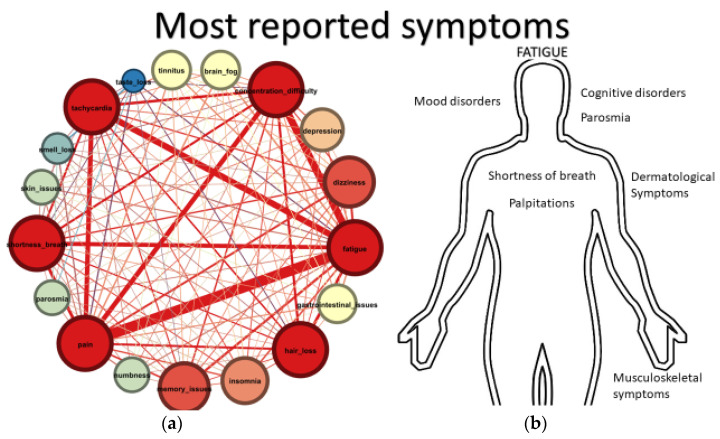
(**a**) Network Analysis graph of persistent symptoms. Larger nodes imply higher degree centrality. (**b**) Most frequently reported symptoms in Greek Long COVID society (LCGr).

**Figure 4 jpm-12-00987-f004:**
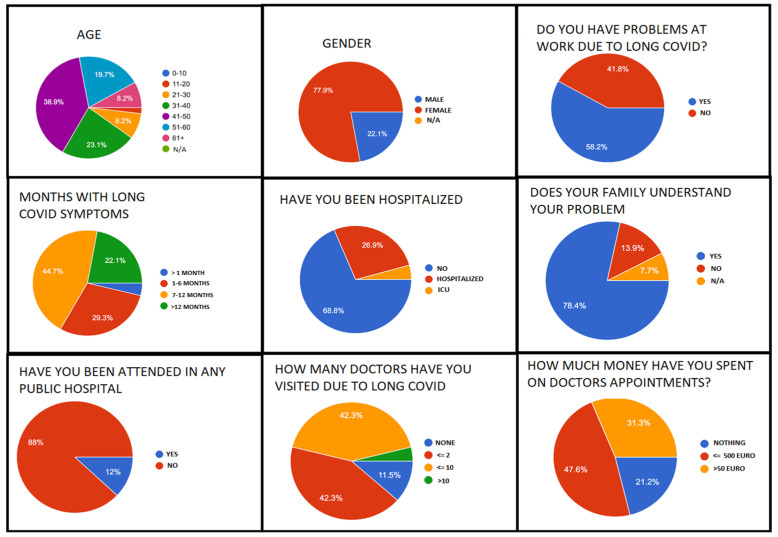
Socioeconomic impact of Long COVID. ICU: Intensive care unit, N/A: Non available.

**Figure 5 jpm-12-00987-f005:**
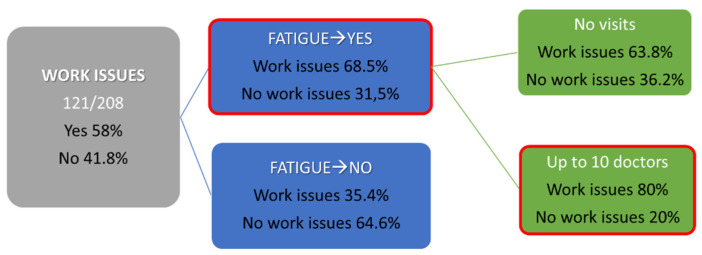
Decision tree reveals that fatigue appears to be muchhigher in patients with work problems.

**Figure 6 jpm-12-00987-f006:**
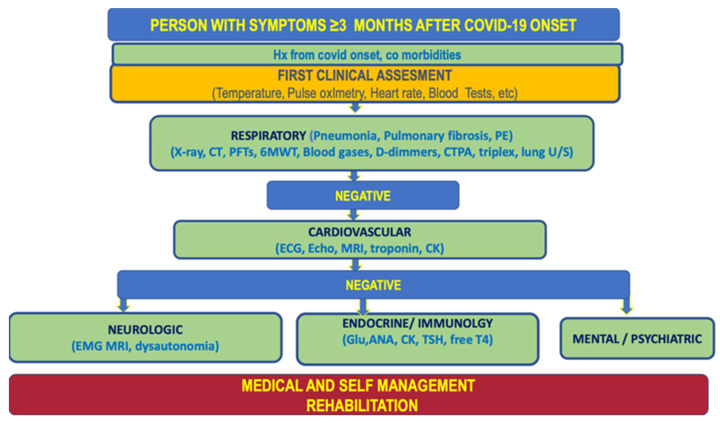
Proposed holistic approach diagnostic algorithm. PE: pulmonary embolism; Chest X-ray; CT: Computer Tomography; PFTs: Pulmonary Function Tests, 6MWT: six minute walking test, CTPA: Computer Tomography pulmonary angiogram, lung U/S ultrasound, ECG electrocardiogram, MRI: Magnetic resonance imaging, EMG: electromyogram, Glu: Glucose, ANA: Antinuclear antibodies, CK: creatine kinase, TSH: Thyroid-stimulating hormone Test, freeT4: Free thyroxine test.

**Table 1 jpm-12-00987-t001:** Questionnaire.

1. Please select your age (in years)
2. Sex
3. Elapsed number of months since you experience symptoms after COVID-19 infection
4. Long COVID symptoms (please use a comma to separate each symptom)
5. Have you been hospitalized?
6. Do you have work issues from the prolonged illness due to Long COVID?
7. Do you have understanding and support in your professional life?
-I am self-employed
-I have not returned to my work yet
8. Do you have understanding and support from your family?
9. Are you being monitored in any public hospital?
10. Have you found appropriate medical support and monitoring?
-Yes, in the public sector
-Yes, at private Medical doctors
-Yes, in the public sector and private medical doctors
11. Have you visited several doctors in order to find a solution to your symptoms?
12. Have you encountered medical doctors who were not aware of the Long COVID syndrome?
They have heard about Long COVID but were not fully aware of its symptoms
13. How much (in Euros) have you spent on doctor visits due to Long COVID?
14. Have you had any rehabilitation treatment?

**Table 2 jpm-12-00987-t002:** Comparison results between hospitalized and non-hospitalized Long COVID patients.

	Hospitalized (*n* =64)	Non-Hospitalized (*n* =144)	*p*-Value
Male Female Fatigue	33.84% (22/64)	17.34% (25/144)	0.00677
	64.61% (42/64)	82.6% (119/144)	0.00677
	71.87% (46/64)	69.44% (100/144)	0.72356
Musculoskeletal symptoms Cognitive disorders Mood disorders	64.06% (41/64)	45.80% (66/144)	0.01519
	46.15% (30/64) 18.75% (12/64)	38.19 (55/144) 6.94 (10/144)	0.27820 0.01061
Palpitations Parosmia Olfactory disorders (incl.parosmia) At least one neurological (excl. fatigue)	26.56% (17/64)	28.4% (41/144)	0.77681
	1.56% (1/64)	13.19% (19/144)	0.00862
	3.13% (2/64)	22.22% (32/144)	0.00058
	75.00% (48/64)	77.77% (112/144)	0.66076
Shortness of breath Dermatological	23.43% (15/64)	15.27% (22/144)	0.15552
	18.75% (12/64)	18.05% (26/144)	0.90478

## Data Availability

The data presented in this study are available on request from the corresponding author.

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
