# Peer review of "The Greek Collaborative Long COVID Study: Non-Hospitalized and Hospitalized Patients Share Similar Symptom Patterns"

_jpm, 2022, doi:10.3390/jpm12060987_

Round 1
Reviewer 1 Report
In this paper Katsarou and colleagues described symptoms and interactions with the social and medical system of 208 Greek patients suffering of Long COVID. The dates were gathered with an online anonymized questionnaire. The study found that there is no difference in the majority of the symptoms between COVID-19 patients who were treated at home and hospitalized patients. Finally, the authors suggested a holistic algorithm to approach patients suffering of protracted COVID-19 symptoms.
The paper brings some novel insights into the Long COVID syndrome in the Greek population. However, some bias and some issues should be highlighted and addressed before considering the publication of the paper, that cannot be published on this form.
1- The major issue for this paper is a selection bias in the population: firstly, the population involved is partially selected by starting in a Facebook page of patients already gathered under the name of “Long COVID Greece patients”. This could have led to a selection of patients already more conscious of the disease or with interpersonal interactions changing the results of the questionnaire. Secondly, using an internet platform could lead to an age selection bias (as stated also by the authors). These points should be evaluated and at least added as limitation of the study.
2-The authors based all their results and conclusions on the assumption of the diagnosis of Long COVID in the study population; however, this was a self-made diagnosis, not confirmed by any physician. This partially undermines the results of the study, especially because the symptoms reported were not evaluated to exclude any other underling disease, maybe not diagnosed. This point should be carefully evaluated and at least added in the limitations of the study.
3- The introduction is surely complete; however, it should be more focused on the topic of the research. In the effort to completely cover the literature on Long COVID this section of the article feels a little confusing and overwhelming. I ask the authors to shorten the section and to focus it on the definition of Long COVID in adults, to be consistent with the study population.
There are also some minor points that should be addressed to improve the paper:
1-The terminology for COVID-19 and other related terms should be consistent through the paper and with the current scientific terminology (e.g., COVID-19 should be always spelled all majuscule, Long COVID should be spelled with the capital L and COVID all majuscule, etc.).
2-Table 1 and its title should be consistent with each other; the title reports “hospitalized and non-hospitalized” patients, the table has a “post mild acute phase” column. Furthermore, in this table a column with dates of all the population could be useful.
3- An English translation of the questionnaire could be added as a supplement of the article to give the possibility of reproducing the study.
Author Response
Dear Editor,
Thank you for your communication regarding the aforementioned manuscript submission. Below is a point-by-point response to the reviewer’s comments. First, we would like to thank the reviewers for the encouraging comments and suggestions to improve our paper. We have addressed the suggestions point by point according to the instructions and have made the appropriate changes in the manuscript. We believe that the manuscript has improved and hope that it merits publication in the Special Issue "Personalized Medicine for Covid-19 Patients-Clinical Considerations" of JCM.
RESPONSE TO REVIEWER
In this paper Katsarou and colleagues described symptoms and interactions with the social and medical system of 208 Greek patients suffering of Long COVID. The dates were gathered with an online anonymized questionnaire. The study found that there is no difference in the majority of the symptoms between COVID-19 patients who were treated at home and hospitalized patients. Finally, the authors suggested a holistic algorithm to approach patients suffering of protracted COVID-19 symptoms.
The paper brings some novel insights into the Long COVID syndrome in the Greek population. However, some biases and some issues should be highlighted and addressed before considering the publication of the paper, which cannot be published on this form.
1- The major issue for this paper is a selection bias in the population: firstly, the population involved is partially selected by starting in a Facebook page of patients already gathered under the name of “Long COVID Greece patients”. This could have led to a selection of patients already more conscious of the disease or with interpersonal interactions changing the results of the questionnaire. Secondly, using an internet platform could lead to an age selection bias (as stated also by the authors). These points should be evaluated and at least added as a limitation of the study.
Response: Thank you for bringing this to our attention. We have included these issues as limitations of the study.
2- The authors based all their results and conclusions on the assumption of the diagnosis of Long COVID in the study population; however, this was a self-made diagnosis, not confirmed by any physician. This partially undermines the results of the study, especially because the symptoms reported were not evaluated to exclude any other underling disease, that maybe not diagnosed. This point should be carefully evaluated and at least added in the limitations of the study.
Response: Thank you for this comment. Indeed, patients with mild disease did not had access to a medical facility. However, we believe that since we used free texting in order to record symptoms, patients were not biased but spontaneously expressed their symptoms. Finally, the study did not address any diagnoses but symptoms.
3- The introduction is surely complete; however, it should be more focused on the topic of the research. In the effort to completely cover the literature on Long COVID this section of the article feels a little confusing and overwhelming. I ask the authors to shorten the section and to focus it on the definition of Long COVID in adults, to be consistent with the study population.
Response: Thank you for this comment; we have edited the text as per the reviewer’s suggestion. Specifically, we have focused the introduction on the definition of Long COVID in adults in order to be consistent with the study population and shortened the introduction
There are also some minor points that should be addressed to improve the paper:
1-The terminology for COVID-19 and other related terms should be consistent throughout the paper and with the current scientific terminology (e.g., COVID-19 should be always spelled all majuscule, Long COVID should be spelled with the capital L and COVID all majuscule, etc.).
Response: We thank the reviewer for this comment. We checked the manuscript and corrected all terms used so as to be consistent (COVID-19, Long COVID, etc).
2-Table 1 and its title should be consistent with each other; the title reports “hospitalized and non-hospitalized” patients and the table has a “post mild acute phase” column. Furthermore, in this table, a column with dates of all the population could be useful.
Response: We thank the reviewer for this comment. We acknowledge this inconsistency in the title and corrected it to read:
Table 2 Comparison results between hospitalized and non-hospitalized Long COVID patients
|
Hospitalized (n=64) |
Non-hospitalized (n=144) |
Pvalue |
Male Female Fatigue |
33.84% (22/64) |
17,34% (25/144) |
0.00677 |
64,61% (42/64) |
82.6% (119/144) |
0.00677 |
|
71,87% (46/64) |
69,44% (100/144) |
0.72356 |
|
Musculoskeletal symptoms Cognitive disorders Mood disorders |
64,06 %(41/64) |
45,80% (66/144) |
0.01519 |
46,15 %(30/64) 18,75 % (12/64) |
38,19 (55/144) 6,94 (10/144) |
0.27820 0.01061 |
|
Palpitations (mention as tachycardia) Parosmia Olfactory disorders (incl.parosmia) At least one neurological (excl. fatigue) |
26,56% (17/64) |
28,4%(41/144) |
0.77681 |
1,56% (1/64) |
13.19%(19/144) |
0.00862 |
|
3,13% (2/64) |
22.22% (32/144) |
0.00058 |
|
75,00%(48/64) |
77.77%(112/144) |
0.66076 |
|
Shortness of breath Dermatological |
23,43% (15/64) |
15,27% (22/144) |
0.15552 |
18,75% (12/64) |
18,05% (26/144) |
0.90478 |
3.1.1. Neurological, cognitive, and psychiatric symptoms
3- An English translation of the questionnaire could be added as a supplement to the article to give the possibility of reproducing the study.
Response: Thank you for this comment. We added the questionnaire used in Table 1.
Table 1 Questionnaire |
1. Please select your age (in years) 2. Gender 3. Number of months passed since you were diagnosed with Long COVID 4. Long COVID symptoms (please use a comma to separate each symptom) 5. Have you been hospitalized? 6. Do you have work issues from the prolonged illness due to Long COVID? 7. Do you have understanding and support in your professional life? -I am self-employed -I have not returned to my work yet 8. Do you have understanding and support from your family? 9. Are you being followed up in any public hospital? 10. Have you found appropriate medical support and monitoring? -Yes in the public sector -Yes at private Medical doctors -Yes, in the public sector and private medical doctors 11. Have you visited several doctors in order to find a solution to your symptoms? 12. Have you encountered medical doctors who were not aware of the Long COVID syndrome? They have heard about Long COVID but were not fully aware of its symptoms 13. How much (in Euros) have you spent on doctor visits due to Long COVID? 14. Have you had any rehabilitation treatment? |
|
- Results: fatigue is included under 'Neurological' symptoms. I am not sure this is appropriate. 'Tachycardia' is not a symptom; the correct term is palpitation.
Response: We thank the reviewer for this comment. We corrected the term tachycardia and used palpitation which is a symptom. Since fatigue could be attributed to various underlying causes we reported it separately throughout the text.
- Rehabilitation is hardly mentioned in the results. In addition, the section in the discussion does not provide suggestions for components of a rehabilitation program for patients with long COVID. I suggest either this section is developed further, or the mention of rehabilitation is removed from the title.
Response: We thank the reviewer for this comment. Indeed we removed rehabilitation from the new title which is “The Greek Collaborative Long COVID Study: Non-hospitalized and hospitalized patients share similar symptom patterns”
The authors need to revise the references list as some are not written correctly such as reference1.
Response: We thank the reviewer for this comment. We have completed a full review of the paper to ensure references selected to support statements in the paper are more appropriately selected and checked all references to be written correctly, using end note.
Again we thank the reviewers for their valuable comments and suggestions which we believe have clarified and improved the manuscript. We hereby submit the revised version as a clean updated manuscript and hope that it merits publication in Journal of Clinical Medicine
We thank you for your consideration,
Kind regards,
Paraskevi Katsaounou
Associate Professor of Respiratory Medicine
Medical School National and Kapodistrian University of Athens
First ICU Clinic, Evangelismos Hospital, Athens, Greece
Reviewer 2 Report
The authors give the results from a questionnaire sent to members of a social network group who self-identify as having long COVID. The topic is of interest, but the manuscript is not easy to read and contains a lot of details that are not relevant to the main subject of the manuscript which is clinical/symptom-oriented. My specific comments are as follows:
- Title: do the authors think 'long haulers' the best descriptive to use in the title? I suggest using a more familiar term.
- Introduction: the word length of this section is >2000 words which is very unusual. There is a lot of repetition, and much of the details could among putative mechanisms could be removed and only succinctly summarised in the discussion. I suggest the authors can move all these details into a review article they could submit separately.
- Methods: the authors should include a copy of the questionnaire that was sent out to respondents
- Results: fatigue is included under 'Neurological' symptoms. I am not sure this is appropriate. 'Tachycardia' is not a symptom; the correct term is palpitation.
- Rehabilitation is hardly mentioned in the results. In addition, the section in the discussion does not provide suggestions for components of a rehabilitation programme for patients with long COVID. I suggest either this section is developed further, or the mention of rehabilitation is removed from the title.
- The authors need to revise the references list as some are not written correctly such as reference1.
Author Response
Dear Editor,
Thank you for your communication regarding the aforementioned manuscript submission. Below is a point-by-point response to the reviewer’s comments. First, we would like to thank the reviewers for the encouraging comments and suggestions to improve our paper. We have addressed the suggestions point by point according to the instructions and have made the appropriate changes in the manuscript. We believe that the manuscript has improved and hope that it merits publication in the Special Issue "Personalized Medicine for Covid-19 Patients-Clinical Considerations" of JCM.
RESPONSE TO REVIEWER
Τhe authors give the results from a questionnaire sent to members of a social network group who self-identify as having long COVID. The topic is of interest, but the manuscript is not easy to read and contains a lot of details that are not relevant to the main subject of the manuscript which is clinical/symptom-oriented. My specific comments are as follows:
- Title: do the authors think 'long haulers' the best descriptive to use in the title? I suggest using a more familiar term.
Response: We thank the reviewer for this comment. In paragraph 3 we explain the terminology of 'long haulers' (https://www.ncbi.nlm.nih.gov/pmc/articles/PMC7539940/). However, we decided to shorten the title and change the title using a more familiar term as advised. The revised title is “The Greek Collaborative Long COVID Study: Non-hospitalized and hospitalized patients share similar symptom patterns”.
- Introduction: the word length of this section is >2000 words which is very unusual. There is a lot of repetition, and much of the details could among putative mechanisms could be removed and only succinctly summarized in the discussion. I suggest the authors can move all these details into a review article they could submit separately.
Response: We thank the reviewer for this comment. We rewrote and shortened the introduction
- Methods: the authors should include a copy of the questionnaire that was sent out to respondents.
Response: We thank the reviewer for this comment. We added the questionnaire used in Table 1.
Table 1 Questionnaire |
1. Please select your age (in years) 2. Gender 3. Number of months passed since you were diagnosed with Long COVID 4. Long COVID symptoms (please use a comma to separate each symptom) 5. Have you been hospitalized? 6. Do you have work issues from the prolonged illness due to Long COVID? 7. Do you have understanding and support in your professional life? -I am self-employed -I have not returned to my work yet 8. Do you have understanding and support from your family? 9. Are you being followed up in any public hospital? 10. Have you found appropriate medical support and monitoring? -Yes in the public sector -Yes at private Medical doctors -Yes, in the public sector and private medical doctors 11. Have you visited several doctors in order to find a solution to your symptoms? 12. Have you encountered medical doctors who were not aware of the Long COVID syndrome? They have heard about Long COVID but were not fully aware of its symptoms 13. How much (in Euros) have you spent on doctor visits due to Long COVID? 14. Have you had any rehabilitation treatment? |
|
- Results: fatigue is included under 'Neurological' symptoms. I am not sure this is appropriate. 'Tachycardia' is not a symptom; the correct term is palpitation.
Response: We thank the reviewer for this comment. We corrected the term tachycardia and used palpitation which is a symptom. Since fatigue could be attributed to various underlying causes we reported it separately throughout the text.
- Rehabilitation is hardly mentioned in the results. In addition, the section in the discussion does not provide suggestions for components of a rehabilitation program for patients with long COVID. I suggest either this section is developed further, or the mention of rehabilitation is removed from the title.
Response: We thank the reviewer for this comment. Indeed we removed rehabilitation from the new title which is “The Greek Collaborative Long COVID Study: Non-hospitalized and hospitalized patients share similar symptom patterns”
- The authors need to revise the references list as some are not written correctly such as reference1.
Response: We thank the reviewer for this comment. We have completed a full review of the paper to ensure references selected to support statements in the paper are more appropriately selected and checked all references to be written correctly, using end note.
Again we thank the reviewers for their valuable comments and suggestions which we believe have clarified and improved the manuscript. We hereby submit the revised version as a clean updated manuscript and hope that it merits publication in Journal of Clinical Medicine
We thank you for your consideration,
Kind regards,
Paraskevi Katsaounou
Associate Professor of Respiratory Medicine
Medical School National and Kapodistrian University of Athens
First ICU Clinic, Evangelismos Hospital, Athens, Greece
Reviewer 3 Report
Dear Authors,
Your study has important findings. But it is hard to read and undersand and also it is too long. You should revise whole manuscript for grammary, language and length. It looks like e review not a study.
In the abstract: you should write the titles of the parts: “Aim, Methods, Results, Conclussion”.
In Introduction part: In line 51 you wrote first “PASC”, later “PACS” two times, you should correct this sentence. Introduction part is too long. It is very difficult to read and understand. It goes on 3 pages, it must be half or maximum 1 page and include 3-4 paragraphs. You should rewrite introduction. You should write first descriptions about postacute covid, long COVÄ°D of WHO. Later you should write the most important results of the studies about this subject. At the end of the introduction you should write your primary aim as clearly. You had written a lot of aims.
In the methods: You should use a flow chart about participants. How many patients did you scan? You should write exclusion criteria. You should write more detailed about questionnaire and questions. How many parts and questions does it include? You should write statistical methods which you used.
The results part is also too long. You should write only your findings. You should not write your comments as you wrote in the 3.1.3 and 3.1.4 paragraphs.
In the discussion: You should write your most important findings and the importance of your study in the first paragraph. After you should discuss your findings with literature. In conclussion part you should write your last message clearly.
In the References: There are many references. You should exclude some of them. You should review all references for writing rules of the journal. in the first reference you should exclude “Author 1, A.B.; Author 2, C.D. Title of the article. Abbreviated Journal NameYear, Volume, page range”. References 7, 11, 12, 28, 40, 50 could not be found on-line. You should write author names of 43th reference.
Round 2
Reviewer 1 Report
Firstly, I thank the authors for the work and the answers given. The paper is now clearer, and especially the introduction is now easier to read. However, a major issue remains at the core of the paper that has not been addressed and creates a bias to the conclusions of the paper itself.
1- The main remaining issue is that the symptoms are self-reported by the patients, with a majority of them that never (53.4% are not attended by medical staff). Those symptoms may be caused by other diseases and could be not related to Long COVID but to other medical conditions not known by the participants. As stated by the authors all these patients should be evaluated step by step to exclude other conditions misinterpreted as Long-COVID.
Therefore, I agree that free-text recorded symptoms could reduce the bias, but I do not believe that it reduces the possibility of misinterpreted symptoms.
Another issue that I found during this second revision is:
1- references 51-52 are self-citations that feel forced in the context of the paper, and the conclusion of the paper itself seems driven by the need of this citations. Surely genetic patters could lead to different patterns of symptoms, and different long-term response to the SARS-CoV-2 infection, however those citation are not necessary to affirm this.
There is also a minor point that should be addressed to improve the paper:
1-The terminology for COVID-19 and other related terms should be revised another time, in some cases it has not been corrected properly.
2- The reference list and the order of the references in the paper should be revised (in the introduction the order goes from [4] to [10-11], than to [6] without a citation [5]).
Author Response
Dear reviewer,
We thank you for the comments that hleped us improve our manuscript.
We revised the references and the terminology for COVID-19 and other related terms as advised.
As indeed the symptoms are self-reported by the patients,this is pointed as a limitatation of our study. However, this is how globally long covid is defined. We tried to reduce the bias using free-text recorded symptoms and currently we are examining in our multidisplinary outpatient clinic these long covid patients in order to provide a second manuscript with the results.
Reviewer 2 Report
I thank authors for addressing my comments and undergoing multiple changes to the manuscript. I have no further comments to make.
Author Response
We thank you again for your comments that helped us improve our manuscript
Reviewer 3 Report
Dear Authors,
You revised very well the manuscript. But it needs still some revision.
In the Introduction: You should correct the numbers of the references. You should shorten the introduction, it is still too long. You can exclude the findindigs of the autopsy studies, because you don’t have any autopsy findings in your study.
In the Materails and methods: In the flow chart, you included 208 cases but you wrote 161 female and 46 male ????, you should correct.
In the results: You should control the percentage levels in the line 194-195, because the total of them is not 100%.
You should correct the number of all references in the whole main text.
Author Response
We thank you again for your comments that helped us improve our manuscript.
We further reduced the introduction and revised the references as advised. We also corrected the number of cases in our flow chart and controled the percentage levels.